# Genetic Interactions of Awnness Genes in Barley

**DOI:** 10.3390/genes12040606

**Published:** 2021-04-20

**Authors:** Biguang Huang, Weiren Wu, Zonglie Hong

**Affiliations:** 1Key Laboratory for Genetics, Breeding and Multiple Utilization of Crops, Ministry of Education, Fujian Agriculture and Forestry University, Fuzhou 350002, China; hbg1989@163.com; 2Fujian Collegiate Key Laboratory of Applied Plant Genetics, Fujian Agriculture and Forestry University, Fuzhou 350002, China; 3Department of Plant Sciences, University of Idaho, Moscow, ID 83844, USA; 4Fujian Key Laboratory of Crop Breeding by Design, Fujian Agriculture and Forestry University, Fuzhou 350002, China

**Keywords:** barley, awn, gene interaction, gene mapping, transcription factor

## Abstract

Awns are extending structures from lemmas in grasses and are very active in photosynthesis, contributing directly to the filling of the developing grain. Barley (*Hordeum vulgare* L.) awns are highly diverse in shape and length and are known to be controlled by multiple awn-related genes. The genetic effects of these genes on awn diversity and development in barley are multiplexed and include complementary effect, cumulative effect, duplicate effect, recessive epistasis, dominant epistasis, and inhibiting effect, each giving a unique modified Mendelian ratio of segregation. The complexity of gene interactions contributes to the awn diversity in barley. Excessive gene interactions create a challenging task for genetic mapping and specific strategies have to be developed for mapping genes with specific interactive effects. Awn gene interactions can occur at different levels of gene expression, from the transcription factor-mediated gene transcription to the regulation of enzymes and metabolic pathways. A better understanding of gene interactions will greatly facilitate deciphering the genetic mechanisms underlying barley awn diversity and development.

## 1. Background

Barley (*Hordeum vulgare* L.) is one of the oldest domesticated grain crops and has been cultivated for over 8000 years. With an annual production of 150 million tones worldwide, barley is ranked fourth in quantity produced among grains behind maize, rice, and wheat. Different from rice and wheat, most barley cultivars develop awns. Awns are extending structures from lemmas in grasses and are very active in photosynthesis, contributing directly to the filling of the developing grain. In addition, awns can protect against animal predation before grain harvest and help disperse dry seeds in the wild and anchor germinating seeds to the soil [1].

In a living cell of higher organisms, thousands of genes are expressed simultaneously and a specific trait is often determined or influenced by more than one genetic locus. Thus, gene interaction is very common in genetic studies. Interactions among genes frequently lead to distortion of simple Mendelian segregation ratios, even producing novel phenotypes. For two pairs of independent genes, a segregation ratio of 9:3:3:1 in F_2_ is expected. However, this ratio could be distorted when the two pairs of genes interact with duplicate effect, complementary effect, cumulative effect, dominant epistasis, recessive epistasis, or inhibiting effect [2]. When two dominant genes have the same effect on a trait, indicating duplicate effect, it leads to a segregation ratio of 15:1. When two dominant genes complement each other, the complementation effect produces a segregation ratio of 9:7. A ratio of 9:6:1 is often indicative of the cumulative effect of two dominant genes, while a ratio of 13:3 suggests the inhibiting effect of two dominant genes. When one gene suppresses the effect of another, the epistasis effect would be observed, with a ratio of 12:3:1 in dominant epistasis and 9:3:4 in recessive epistasis [2].

The effect of gene interaction is often observed in plants [3,4,5,6,7]. In wheat (*Triticum aestivum* L.), purple grain is controlled by two complementary dominant genes; waxiness of grain endosperm by three duplicate recessive genes [8]; and awn development by the interactions among three dominant awnless genes, *B1*, *B2*, and *Hd* (hooded) [9]. In rice (*Oryza sativa* L.), male fertility is determined by two duplicate genes [10], the color of exoceomum by two complementary genes, and awn development by *An-1* and *An-2* with additive effect [11,12,13].

Modern genetic studies have revealed that genes usually function in networks to control trait development. Thus, it is important to characterize the function of genes in the whole context of gene interaction. Development of near isogenic lines (NILs) is a strategy for distinguishing the function of individual genes [14]. In barley, there has been a world-wide effort since 1985 to introduce various mutated genes into a common genetic background, the cultivar Bowman, to produce specific NILs [15,16,17]. Some of the NILs carry genes affecting morphological and developmental processes, such as the floral bract gene *Hooded lemma 1* (*Kap1*) and the spike row-type gene *Six-rowed spike 1* (*Vrs1*) [18,19]. As different mutant genes carried by the NILs function in the same cultivar, the effect of genetic background variation can be eliminated. This will enable more precise comparison of gene effects and, therefore, more reliable analysis of gene interactions. A number of different genes have been identified for awn length development in barley NILs, such as *lks5*, *lks9*, and *lks11,* and have been shown to play very important roles in awn development [16]. This review will be focused on the advances in genetic studies of barley awnness gene interaction types, specific gene mapping strategies, and gene interaction mechanisms.

## 2. Interactions of Awnness Genes in Barley

The development of grass awns has become a topic of intense research largely owing to its important function in grain filling [1]. Awn in barley (*Hordeum vulgare* L.) is more diverse than in rice and wheat [20]. The awnness trait can be distinguished into different phenotypes, including awnless, straight (long, longer than the length of spike axis; short, shorter than the length of spike axis; and awnlet, shorter than 1 cm), hooded, leafy, and crooked at the end of a lemma (Figure 1). They are controlled by different awnness loci that interact with each other. Analysis of the interactions among these loci can potentially reveal the mechanisms by which awn development is regulated. This topic has not been systematically studied, although some historic and recent investigations may have provided interesting hints [21,22,23].

When a hooded-awn line is crossed with a long-awn line, the hooded awn is always dominant over the long awn, and the awn trait is inherited as a single locus independent of the row type gene [24,25]. However, when a hooded-awn line is crossed with an awnless line, the inheritance becomes more complicated. The F_2_ population segregates into hooded and normal awns in a ratio of 9:7, suggesting that the hooded-awn phenotype is controlled by two complementary factors [26]. It has been proposed that two dominant awn genes are necessary for the development of hooded awn, indicating the complementary effect of hooded genes, or the recessive epistasis of normal awn over hooded awn [22]. Analysis has revealed that the recessive short awn gene *lks2* is epistatic over the hooded gene *Kap1* (*K*), giving a ratio of 9:3:4 in F_2_ [23].

A recent study on the interaction of awnness loci has led to the identification of *Lsa1*, a dominant locus underlying the awnlessness on lateral spikelets [21]. Different from the above recessive epistasis of awned over hooded, a hierarchical dominant epistasis of awnless over hooded and awned trait has been identified, and the awnless gene of central row has been shown to be epistatic over the awn trait of lateral row [21]. Interestingly, the awnless gene *B1* in wheat acts as a dominant suppressor of the hooded phenotype [8]. Similarly, barley *Lsa1* also exhibits a dominant epistatic effect over the hooded allele *Kap1.* A candidate of the wheat *B1* homologs in barley has been identified as *HvFT-3*, which has a similar chromosome location as *Lsa1*, implying that *HvFT-3* might be the candidate gene of *Lsa1* [21,27].

The awn phenotype of mutant *leafy lemma* (*lel*) is controlled by two independent and duplicate loci, *lel1* and *lel2* [15], as suggested by analysis of segregation data of a cross between *lel* and the wild type (WT) control, which produces an F_2_ ratio of 15:1 for WT and *lel* awns [28]. The *lel2* locus has been identified as *lks5* on chromosome 4H, while *lel1* is inferred on chromosome 2H [15,16,28]. Awn length determined by two duplicate dominant genes with an F_2_ ratio of 15:1 for long-awned to short-awned has also been reported [29].

Grunewaldt (1974) reported that awnless (*S*) is dominant to awned (*s*) and long awn (*A*) is dominant to short awn (*a*), respectively, and the F_2_ generation of the cross between a short awn mutant and an awnless variety segregates into 13 awnless/short awn versus 3 long awn [30]. He proposed that *S* is an inhibitor of *A*, suppressing the awn development. Therefore, the genotypes *S_A_*, *S_aa*, and *ssaa* show the phenotype of awnless or short awn, while the genotype *ssA_* shows the phenotype of long awn.

The length of awns is a complex trait controlled by the interactions of several genes, including *lr1* (for reduced length of lateral awns), *lks5* (or *lk5*, for short awn 5), and *lks2* (or *lk* and *lk4*, for short awns) [22,31]. Understanding of the specific effect of an awn gene among the interactive loci will help define the awn gene and illustrate its function. *lr1* is similar to *lr* reported by Leonard (1942), both leading to the reduced length of lateral awns, but *Lr* has no effect on the awn length on the central rows and is allelic to the row-type gene *vrs1* [32,33,34]. *Lr1* controls the awn development on the central and lateral rows, having additional effects on the awn length on the lateral rows. The homozygous genotype of *lr1* leads to the awnless phenotype on the lateral rows. *lr1, lks2,* and *lks5* act together to regulate the awn length in a predicted manner [22,31]. The central awn is about half of the full length in the genotype *Lr1_ lks5/lks5* and about one-quarter of the full length in *lr1/lr1 Lks5_*, suggesting that the effect on central awn length of *Lr1* is more prominent than *Lks5*, and the awns are in full length when the two genes are dominant, and awnlet when the two genes are recessive [22]. In the presence of *Lr1*, the two short awn loci, *Lks2* and *Lks5,* act additively to regulate awn length. The F_2_ population is segregated into a ratio of 9 long awn (*Lks2/*_ *Lks5/*_) to 6 short awn (*Lks2/_ lks5/lks5* and *lks2/lks2 Lks5_*) to 1 awnless (*lks2/lks2 lks5/lks5*). The effect of *Lks2* is dependent upon the presence of *Lr1*, because the awn genotypes of both *lr1/lr1 Lks2*_ *Lks5*_ and *lr1/lr1 lks2/lks2 Lks5*_ are indistinguishable, having short awns, indicating *lr1lr1* may have recessive epistasis on the expression of *Lks2* [31].

Complementation test, also known as cis-trans test, is commonly used to determine if the genetic loci of two similar mutants are allelic or non-allelic. In this test, F_1_ plants of two allelic mutants would display mutant phenotype, whereas two non-allelic mutant genes would complement each other, leading to a wild-type phenotype in their F_1_ plants. If non-allelic genes determine the similar traits, then an interaction between them happens. Allelism tests among *int-b.3*, *int-b.6*, *int-b.75,* and *vrs2* indicate that they are allelic to the row type gene *Vrs2* [35]. Complementation tests exclude the allelism between short awned *suK* loci, *lks2*, and *Hooded* (*K*) [36]. Preliminary data from allelism tests indicate that the short-awn mutation *lk2* and two incomplete dominant mutations for short awns in Morex and KM-200, respectively, are allelic, and that the short-awn mutation *lk5* in Morex and a *lk5* mutation in KT4-218 and a mutation designated *lk* are likewise allelic [37].

In short, the genetic control of awn development in barley is very complicated, involving all sorts of gene interactions. Part of the genes and interactions for awn development found in barley are summarized in Table 1.

## 3. Strategies for Mapping Interactive Genes

Genetic mapping and linkage analysis are often complicated by the presence of gene–gene interactions. Special models and methods are required for mapping interactive loci. A strategy for mapping interactive genes in an F_2_ population has been proposed, in which markers linked with one of the interactive loci (say, *A*/*a*) are identified by bulked segregant analysis (BSA) at first, and then the markers are used to genotype a group of segregants of the same genotype(s) at the target locus (either all *A*_or all *aa*) and the genetic distances between the markers and the target locus are estimated by linkage analysis using the software Mapmaker/Exp 3.0 [38]. The core of this mapping strategy is to identify the linked markers of the interactive loci by finding out and comparing the allelic difference between segregant groups in the mapping population. With this mapping strategy, two interactive genes, *prbs* and *vrs1*, between which *prbs* is recessive epistatic over *Vrs1*/*vrs1*, have been mapped on the SSR (Simple Sequence Repeat) linkage map in barley [39,40]. This strategy will be useful for mapping barley awnness genes involving various types of interactions among them. Specific methods for mapping these awnness genes with different interaction situations are described below.

Mapping of *L*/*l* and *H*/*h* with recessive epistasis. Based on awn morphology, the F_2_ population can be distinguished into three segregant groups: hooded awn (*L_H_*), long awn (*L_hh*), and short awn (*llH_* + *llhh*). The linker markers of *L*/*l* locus can be identified by comparing the long awn group or hooded awn group with the short awn group, while those of *H*/*h* locus can be identified by comparing the hooded awn group with the long awn group (Figure 2A). The subsequent marker linkage analysis can be performed using any of the groups for *L*/*l* and the hooded awn or long awn group for *H*/*h*, respectively.

Mapping of *N*/*n* and *H*/*h* with complementary effect. The F_2_ population can be divided into two segregant groups: hooded awn (*N_H_*) and normal awn (3 *N_hh* + 3 *nnH_* + 1 *nnhh*). There are no allelic differences between the two groups. However, the *NNHH* individuals inside the hooded awn group can be identified from their F_3_ families, in which the awnness trait does not segregate. Thus, the linked markers of either *N*/*n* or *H*/*h* locus can be identified by comparing the *NNHH* group with the normal awn group (Figure 3A). The subsequent marker linkage analysis can be performed using the hooded awn group for both loci.

Mapping of *C*/c and *D*/*d* with cumulative effect. There are three segregant groups in F_2_: long awn (*C_D_*), short awn (*C_dd* + *ccD_*), and awnlet (*ccdd*). The linked markers of either *C*/*c* and *D*/*d* locus can be identified by comparing the long awn group or short awn group with the awnlet group (Figure 4A). The subsequent marker linkage analysis can be performed using the long awn or awnlet group for both loci.

Mapping of *N*/*n* and *L*/*l* with duplicate effect. There are two segregant groups in F_2_: normal awn (*N_L_* + *N_ll* + *nnL_*) and leafy awn (*nnll*). The linked markers of either *N*/*n* or *L*/*l* locus can be identified by comparing the two groups (Figure 5A), and the subsequent marker linkage analysis can be performed using the leafy group for both loci.

Mapping of *I*/*i* and *F*/*f* with inhibiting effect. There are two segregant groups: awnless (*I_F_* + *I_ff* + *iiff*) and long awn (*iiF_*). The linked markers of *I*/*i* locus can be identified by comparing the two groups (Figure 6A). For *F*/*f* locus, however, it is necessary to use the F_3_ of the long awn F_2_, in which there are two segregant groups: long awn (*iiF_*) and awnless (*iiff*). Therefore, a comparison can be made between these two F_3_ groups to identify the linked markers of *F*/*f* locus. The subsequent marker linkage analysis can be performed using the long awn group for both loci.

Similar strategies can be applied to more complicated inheritance. For example, in the hierarchical dominant epistasis interactions involving four pairs of genes in barley (Figure 7A), there are four segregant groups: awnless (*A_* + *aaB_*), hooded awn (*aabbH_*), long awn (*aabbhhL_*), and short awn (*aabbhhll*) [21]. The linked markers of loci *L*/*l*, *H*/*h*, and *A*/*a* or *B*/*b* can be identified by comparing the long awn group with the short awn group, hooded awn group with long awn group, and awnless group with hooded awn group, respectively. The subsequent marker linkage analysis can be performed using any of the groups for *A/a*, any except the awnless group for *B/b* and *H/h*, and long awn or short awn group for *L/l*, respectively.

## 4. Possible Models of Awness Gene Interactions at Metabolic Level in Barley

A specific phenotype is often the result of a genotype that is expressed through metabolic pathways. Some specific phenotypes that are determined by gene interactions can be explained at the metabolic level [41].

The recessive epistasis of *lks2* (*l*) over the hooded gene *H*/*h* could be explained as illustrated in the diagram (Figure 2B) [23]. The coexistence of *H* and *L* results in the hooded phenotype. In the recessive mutant *lks2*, which lacks functional Lks2 protein, the LKS2 product (long awn) could not be produced and the plants would only develop short awns regardless of the presence of the H enzyme. Thus, *lks2* acts as a recessive epistatic gene over the *Hooded* (*H*) gene. This would produce an F_2_ segregation ratio of 9 hooded to 3 long awned to 4 short awned (Figure 2A). This metabolic pathway explanation of recessive epistasis has been verified by DNA electrophoresis of short awn *lks2* plants having the *Kap* band of hooded plants [42].

The complementation of two dominant awnness genes results in the hooded phenotype [26], as diagrammed in Figure 3. *N* and *H* are independent awnness genes with a complementary effect. Enzyme N and enzyme H alone produce no new phenotype, maintaining normal awn phenotype. However, when both N and H enzymes are present, their products complement each other, giving rise to the new hooded awn phenotype (Figure 3B). The F_2_ segregation ratio would be 9:7 for hooded versus normal awn (Figure 3A).

The cumulative effect of awnness genes controlling awn length is diagrammed in Figure 4A. *C* and *D* are independent genes with a cumulative effect on awn length [31]. An F_2_ segregation ratio of 9:6:1 for long to short to awnlet would be expected. C or D enzyme alone could convert awnlet to short awn phenotype. When both of them are present, their products are additive, giving rise to the long awn phenotype (Figure 4B).

Gene interaction with a duplicate effect has been observed in the leafy awn mutants, as shown in Figure 5. *N* and *L* are independent genes with a duplicate effect on leafy awn morphology [28]. The presence of either of them will give a dominant phenotype, having normal awns. The N and L enzymes are redundant, and only one of them is sufficient to convert leafy awn substrate to normal awn product (Figure 5B). This type of gene interaction would give rise to an F_2_ segregation ratio of 15:1 for normal awn versus leafy awn (Figure 5A).

Gene interaction with an inhibiting effect is diagrammed in Figure 6. The *I* gene has an inhibiting effect on the dominance of *F* [30]. In the presence of *I*, the dominant effect of *F* is completely suppressed, producing awnless phenotype in F_1_ and an F_2_ ratio of 13:3 for awnless versus long awn (Figure 6A). In this type of gene interaction, I enzyme inhibits the transformation of the awnless substrate, giving awnless phenotype. In the absence of I enzyme, F enzyme will catalyze the conversion of awnless substrate into long awn product. When F enzyme encounters with I emzyme, F enzyme loses its activity, producing awnless phenotype, rather than the long awn phenotype (Figure 6B).

Gene interaction with hierarchical dominant epistasis is shown in Figure 7. In this model, *A*, *B*, *H*, and *L* are independent awnness genes with a dominant epistatic effect. The strength of their epistatic effect is in the order of *A*-*B*-*H*-*L*. The epistatic gene *A* exerts its function on both the central row (CR) and lateral row (LR), whereas *B* acts only on LR. The final awn phenotype is determined by the presence of the epistatic gene with the highest supremacy (Figure 7B) [21]. The expected F_2_ segregation ratio would be 48:12:3:1 for awnless to hooded to long awn to short awn on the central rows, and 240:12:3:1 for awnless to hooded to long awn to short awn on the lateral rows (Figure 7A).

## 5. Molecular Mechanisms of Awnness Gene Interactions in Barley

There has so far been a limited understanding of molecular mechanisms of awn gene interactions. Analysis of the expression patterns of awn genes *An-1* and *An-2* in rice suggests that *An-1* regulates the formation of awn primordia, while *An-2* promotes awn elongation [12]. The upstream genes often regulate the expression of downstream genes. In barley, the mutant gene *prbs* (poly-rowed-and-branched spike) is recessive epistatic over the row type gene *Vrs1*/*vrs1*, and *prbs* may function upstream of *vrs1* [7,39]. Two awnness-specific genes have so far been cloned in barley and they each encode a type of transcription factor. One is *Kap1 for hooded lemma 1,* while the other is *lks2* for *short awn 2*. Barley *lks2* is recessive epistatic over the hooded-awn gene *Kap1*, and may function upstream of *Kap1* [42,43]. *Lks2* encodes a SHORT INTERNODES (SHI)-type transcription factor [43]. The SHI proteins contain two conserved regions, the RING-finger motif and the IGGH domain—the former being implicated in zinc-binding and the latter being required for dimerization and transcription activation [44]. *Kap1* encodes KNOX-type transcription factor. KNOX family is known to regulate the maintenance of the shoot apical meristem and the initiation of lateral organs in plants [17]. Awnless gene *B1* in wheat encodes C_2_H_2_ zinc finger proteins [27], which may act as transcriptional repressors to regulate gene expression in developmental processes such as the formation of flower, seed, and rudimentary glume [45]. Barley *HvFT3,* as the counterpart of awnless gene *B1* in wheat, functions upstream of the row-type genes (*Vrs1*, *Vrs4*, and *Int-c*) [27,46].

Genomics studies could be helpful to reveal molecular mechanisms of gene interactions in the future. CRISPR (clustered regularly interspaced short palindromic repeats) knocking out could be a method to prove gene interaction at molecular level. By knocking out the upstream genes, the downstream genes will be expressed and produce corresponding phenotypes. Time-series transcriptomics data provided abundant proof of metabolic processes [47], then gene interactions at molecular and metabolic level could be proved. The yeast two-hybrid technique is also a good system to find the interactive protein, and then provides direct proof for gene interaction.

## 6. Concluding Remarks

Barley awns are highly diverse in morphology, varying from long, short, awnlet, to awnless in length, and from straight to hooded or crooked in shape. A set of genetic loci associated with the diversity of awns have been identified. Interactions among awn genes contribute to this diversity. Further research on awn gene interactions at the genetic, metabolic, and molecular levels will provide insights into the essence of the interactions and elucidate the mechanisms of awn initiation and development in barley.

## Figures and Tables

**Figure 1 genes-12-00606-f001:**
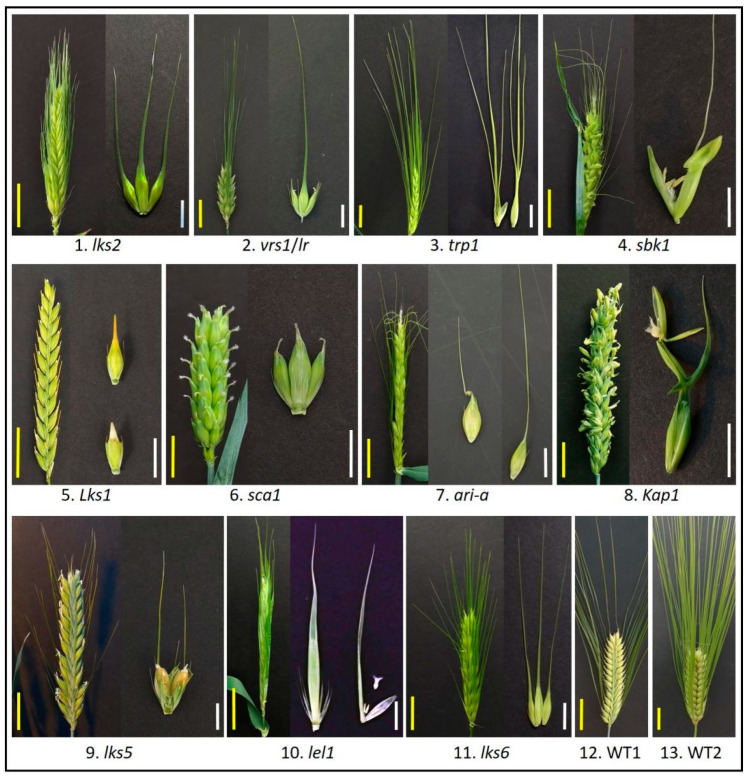
Barley awn diversity. Barley awn mutants obtained from the National Small Grains Collection, USDA-ARS (Agricultural Research Service, United States Department of Agriculture), Aberdeen Idaho. **1**-*lks2*, short awn; **2**-*vrs1*(*lr*), six-rowed with awnless lateral spikelet; **3**-*trp1*, triple awned lemma; **4**-*sbk1*, subjacent hooded lemma; **5**-*Lks1*, awnless; **6**-*sca1*, short crooked awn; **7**-*ari-a*, short awn; **8**-*Kap1*, hooded lemma; **9**-*lks5*, short awn; **10**-*lel1*, leafy lemma; **11**-*lks6*, short awn; **12**-WT1, wild type, two-rowed with long awn; **13**-WT2, wild type, six-rowed with long awn. Yellow scale bar = 3 cm; White scale bar = 1 cm.

**Figure 2 genes-12-00606-f002:**
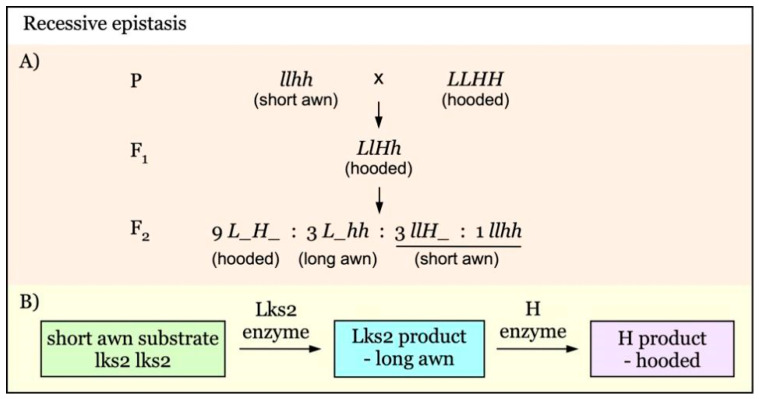
Model of interaction of awnness genes with recessive epistatic effect. (**A**). The short awn gene *lks2* (*l*) is a recessive epistatic gene over the *Hooded* locus (*H*/*h*). The hooded awn phenotype is the result of coexistence of both *L* and *H*. At the lack of *H* gene (*hh*), *Lks2* gives the long awn phenotype. (**B**). In the absence of Lks2 enzyme, there would not be LKS2 product and only short awns are produced regardless of the presence of H. Thus, the recessive *lks2* acts as a recessive epistatic gene that suppresses the effect of the *Hooded* (*H* or *h*) gene.

**Figure 3 genes-12-00606-f003:**
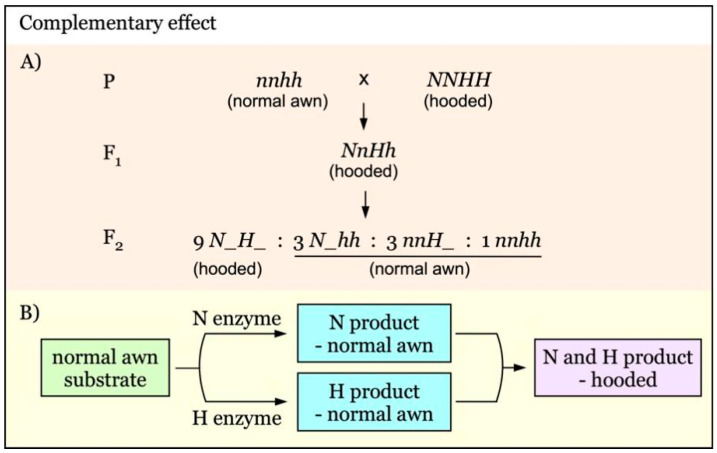
Model of interaction of awnness genes with a complementary effect. (**A**). *N* and *H* are independent genes with a complementary effect. (**B**). In the presence of either N enzyme or H enzyme, only normal awns are produced. In the coexistence of H enzyme and N enzymes, the normal awn intermediate would be further converted to the hooded product (Hooded phenotype).

**Figure 4 genes-12-00606-f004:**
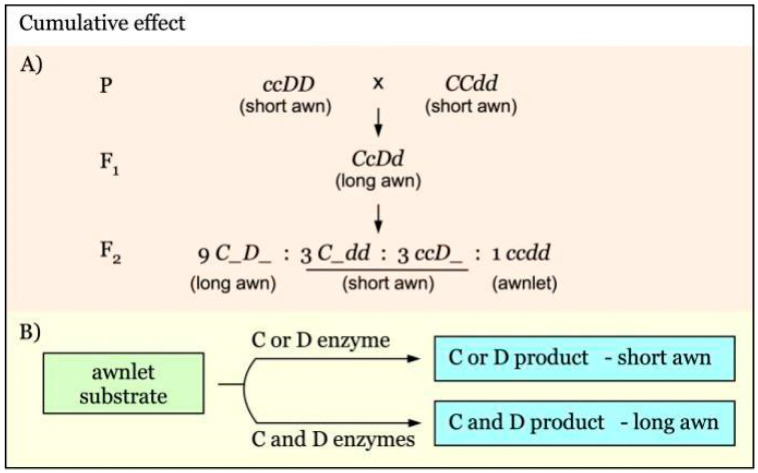
Model of interaction of awnness genes with a cumulative effect. (**A**). *C* and *D* are independent awnness genes that act additively to regulate awn length. (**B**). In the presence of either C or D product, only short awns are developed from the awnlet substrate. The long awn phenotype is produced only with the co-existence of both C and D enzymes, indicating an additive effect.

**Figure 5 genes-12-00606-f005:**
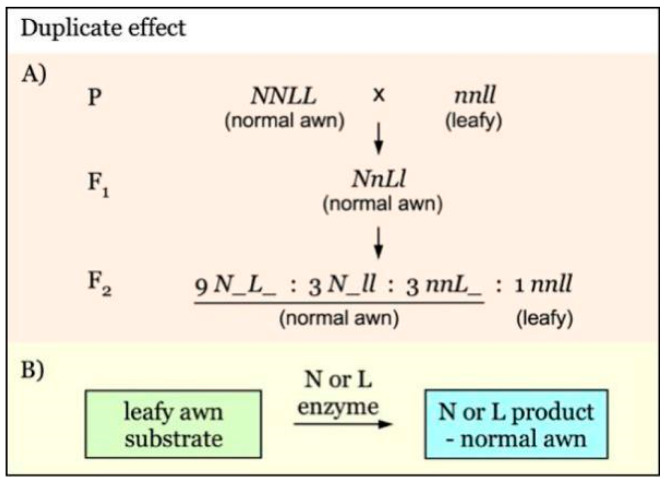
Model of interaction of awnness genes with a duplicate effect. (**A**). *N* and *L* are independent awnness genes with a duplicate effect. The presence of either *N* or *L* is sufficient to result in the formation of normal awns. In the absence of both genes (*nnll*), leafy awns are developed. (**B**). N and L enzymes have a redundant function and only one of them is required for converting the leafy awn substrate to the normal awn phenotype.

**Figure 6 genes-12-00606-f006:**
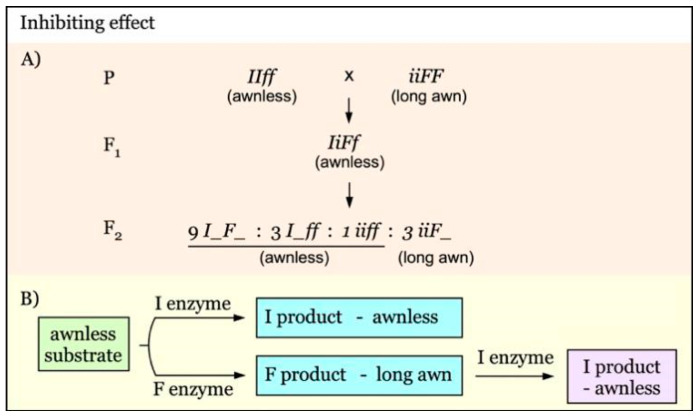
Model of interaction of awnness genes with an inhibiting effect. (**A**). Awnness gene *I* inhibits the dominant expression of the *F* gene. (**B**). In the presence of I enzyme, the conversion of the awnless substrate is inhibited, giving awnless phenotype. In the lack of I enzyme, F enzyme will catalyze the awnless substrate into long awn product. When both F and I enzymes are present, I could inhibit F enzyme, resulting in the awnless phenotype.

**Figure 7 genes-12-00606-f007:**
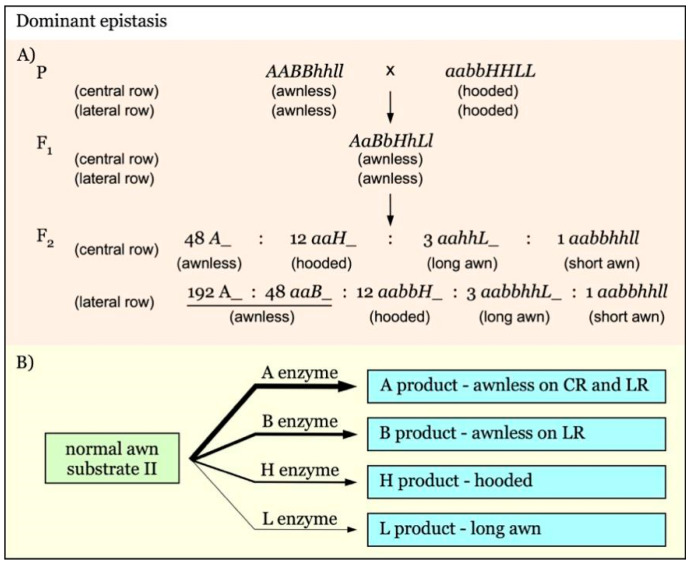
Model of interaction of awnness genes with multiple dominant epistatic effects. (**A**). *A*, *B*, and *H* are dominant epistatic genes over *L*. The strength of their epistatic effect is in the order of *A*-*B*-*H*. The *A* gene functions on both the central row (CR) and lateral row (LR). The *B* gene exerts its epistatic effect only on LR. (**B**). The final awn phenotype is determined by the supremacy of the epistatic gene. The thicker arrows indicate genes with a stronger epistatic effect.

**Table 1 genes-12-00606-t001:** Gene interactions for awn development in barley.

Interaction	Genes	Segregation in F_2_ generation	Reference
Comp.	*N*/*n* and *H*/*h*	9 HA (*N_H_*): 7 NA (*N_hh* + *nnH_* + *nnhh*)	[26]
Dup.	*Lel1*/*lel1* and *Lel2*/*lel2*	15 NA (*Lel1_Lel2_* + *Lel1_lel2lel2* + *lel1lel1Lel2_*): 1 LF (*lel1lel1lel2lel2*)	[28]
Cum.	*Lks2/lks2* and *Lks5/lks5*	9 LA (*Lks2*_ *Lks5*_): 6 SA (*Lks2_ lks5lks5* +*lks2lks2 Lks5_*): 1 AL (*lks2lks2 lks5lks5*)	[31]
Rec. epi.	*Lks2*/*lls2* and *Kap1*/*kap1*	9 HA (*Lls2*_*Kap1*_): 3 LA (*Lks2_kap1kap1*): 4 SA (*lks2lks2Kap1_* + *lks2lks2kap1kap1*)	[23]
Dom. epi.	*Lsa1*/*lsa1* and *Kap1*/*kap1*	12 AL (*Lsa1*_*Kap1*_ + *Lsa1*_*kap1kap1*): 3 HA (*lsa1 lsa1Kap1*_): 1 ST (*lsa1 lsa1kap1kap1*)	[21]
Inh.	S/s and *A*/*a*	13 AL (*S_A_* + *S_aa* + *ssaa*): 3 LA (*ssA_*)	[30]

Note: Comp., complementary; Dup., duplicate; Cum., cumulative; Rec. epi., recessive epistasis; Dom. epi., dominant epistasis; Inh., inhibiting; HA, hooded awn; NA, normal awn; LF, leafy awn; LA, long awn; SA, short awn; AL, awnless; ST, straight awn.

## Data Availability

Not applicable.

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
