# Peer review of "Genetic Interactions of Awnness Genes in Barley"

_genes, 2021, doi:10.3390/genes12040606_

Round 1
Reviewer 1 Report
Manuscript ID: genes-1164102
This review article introduced updated information and current knowledge about awnness phenotype in barley in a classical genetic perspective. The authors did a good job going back in time to review the myths behind barley awnness phenotypes, as well as providing the cutting-edge discoveries of genes regulating barley awns diversity and development. I appreciate the effort of the authors and I really enjoyed reading the manuscript. Here are some minor suggestions and/or comments to the authors:
- I would suggest adding some more information in the background regarding: (1) importance of awns in barley; (2) known molecular mechanisms regulating awn development (known genes, what kind of proteins they encode, etc.); and (3) any known relationship of awn or awnless phenotypes with other important agronomic traits: yield, disease resistance, etc.
- Correct me if I’m wrong, but I think the length of awns in barley is a quantitative trait, right? Please define “long” vs “short” awns before the authors start talking about the genetic studies.
- For section 4, I don’t think “metabolic mechanism” is a proper description of what the authors showed here. Basically the authors presents possible models to explain the genetic interaction, the hypothetical “H enzyme”, “N enzyme” etc. are not confirmed or identified by any molecular biological studies, right? So it’s just hypotheses and probable working models at this stage.
- In section 5, the authors tried to introduce known molecular mechanism of important genes regulating awn development, but probably limited by space and time, the authors did not explain them well, and they tried too hard to include too much information that are not directly related to what they talked about throughout the article. I suggest being less greedy to talk about everything, and move this part into the background information and pick just a few genes to describe in further details, and discard the rest of them.
- For section 6 (there are 2 sections “6”s, by the way), the “outstanding questions” part, I think it is not a good way of presentation as pullet points in a review paper, and I don’t most of these questions are directly related to this article. I suggest deleting this part; or, if the authors are keen on pointing out some more questions that they were not able to cover in this article, I would suggest develop on a couple of these questions in the conclusion section or in a separate paragraph before that.
- For figure 1, I suggest adding a picture of a wildtype cultivar, and maybe a wildtype cultivar or landrace for each of two row and six row phenotypes. And if possible, I would suggest the authors to add a legend in each figure panel to show the dimension of each spike and spikelet that they show in the picture. Although the sizes of spikes and spikelets are not hugely different, we cannot assume all of the readers know that before reading this article. But if the authors don’t have a scale in the pictures when they took the photos, I can definitely understand, and it is ok not to include legends in the photos, but instead, the authors should point it out in the texts, saying that these mutations of awns and row-type phenotype have any impact on spikes/spikelets sizes or not.
Reviewer 2 Report
In this manuscript, Huang et al. gave a comprehensive review of interactions of awn genes in barley and listed several strategies for mapping interactive genes under different situations. Overall, the models of interactions are well presented (figure 2-7) in this paper, and the author also included paragraphs to well explain different interactions through metabolic mechanisms. The paper structure is also well organized. And I think this paper will provide valuable knowledge to other researchers working on awnness gene interactions in barley.
I have a few comments and suggestions to the authors.
- I suggest the authors to have a table that shows the list of awnness gene interactions in barley, with columns of interaction types, phenotypes (e.g., hooded awn vs. long awn), loci involved, segregating ratio. Thus, readers can quickly get information from the table.
- I think maybe it is better to move the paragraph describing the complementation test to somewhere else, like in section 3 (strategies).
- In the first paragraph of section 3 (strategies), the authors mentioned the software Mapmaker/Exp 3.0 and the core of mapping strategy is to find out and compare the allelic difference between segregant groups in the mapping populations. Could the author be more specific about how the algorithm works to compare the allelic difference?
- In section 4 (molecular mechanisms), the authors had a short paragraph that discusses the limited studies on molecular mechanisms of awnness gene interactions. I wonder could the authors have some discussions on how they think genomics studies (e.g., time-series transcriptomics data) could be helpful to reveal molecular mechanisms of gene interactions in future.
- Please rephrase the first sentence of paragraph 2 in section 3 (strategies): “Mapping strategy for L/l and H/h with recessive epistasis.“
Round 2
Reviewer 2 Report
The authors have addressed all my questions in their response and the updated version of manuscript.